# Magnetically Assisted Drug Delivery of Topical Eye Drops Maintains Retinal Function In Vivo in Mice

**DOI:** 10.3390/pharmaceutics13101650

**Published:** 2021-10-09

**Authors:** Marco Bassetto, Daniel Ajoy, Florent Poulhes, Cathy Obringer, Aurelie Walter, Nadia Messadeq, Amir Sadeghi, Jooseppi Puranen, Marika Ruponen, Mikko Kettunen, Elisa Toropainen, Arto Urtti, Hélène Dollfus, Olivier Zelphati, Vincent Marion

**Affiliations:** 1OZ Biosciences, Parc Scientifique de Luminy, Case 922, Zone Entreprise, CEDEX 9, 13288 Marseille, France; mbassetto@ozbiosciences.com (M.B.); fpoulhes@ozbisociences.com (F.P.); awalter@ozbiosciences.com (A.W.); 2INSERM, Ciliopathies Modeling and Associated Therapies Group, Laboratoire de Génétique Médicale, UMRS_U1112, Fédération de Médicine Translationelle de Strasbourg, Université de Strasbourg, 67085 Strasbourg, France; daniel.ajoy@unistra.fr (D.A.); c.obringer@unistra.fr (C.O.); dollfus@unistra.fr (H.D.); 3INSERM, Institute of Genetics and Molecular and Cellular Biology (IGBMC), 67640 Illkrich-Graffenstaden, France; nadiame@igbmc.fr; 4School of Pharmacy, Faculty of Health Sciences, University of Eastern Finland, Yliopistonranta 1C, 70211 Kuopio, Finland; amir.sadeghi@uef.fi (A.S.); jooseppi.puranen@uef.fi (J.P.); marika.ruponen@uef.fi (M.R.); elisa.tropainen@uef.fi (E.T.); arto.urtti@uef.fi (A.U.); 5Kuopio Biomedical Imaging Unit, A.I. Virtanen Institute for Molecular Sciences, University of Eastern Finland, Neulaniementie 2, 70150 Kuopio, Finland; mikko.kettunen@uef.fi; 6Laboratoire de Génétique Médicale, UMRS_U1112, Institut de Génétique Médicale d’Alsace, Fédération de Médicine Translationelle de Strasbourg, Hopiaux Universitaires de Strasbourg, Université de Strasbourg, 67085 Strasbourg, France; 7ALMS Therapeutics, Parc d’Innovation, 650 Boulevard Gonthier d’Andernach, 67400 Illkirch-Graffenstaden, France

**Keywords:** magnetic targeting, non-invasive, topical drug delivery, magnetic nanoparticles, retinal degeneration, Bardet Biedl syndrome, unfolded protein response, small drug molecules

## Abstract

Barded-Biedl syndrome (BBS) is a rare genetic disorder with an unmet medical need for retinal degeneration. Small-molecule drugs were previously identified to slow down the apoptosis of photoreceptors in BBS mouse models. Clinical translation was not practical due to the necessity of repetitive invasive intravitreal injections for pediatric populations. Non-invasive methods of retinal drug targeting are a prerequisite for acceptable adaptation to the targeted pediatric patient population. Here, we present the development and functional testing of a non-invasive, topical, magnetically assisted delivery system, harnessing the ability of magnetic nanoparticles (MNPs) to cargo two drugs (guanabenz and valproic acid) with anti-unfolded protein response (UPR) properties towards the retina. Using magnetic resonance imaging (MRI), we showed the MNPs’ presence in the retina of *Bbs* wild-type mice, and their photoreceptor localization was validated using transmission electron microscopy (TEM). Subsequent electroretinogram recordings (ERGs) demonstrated that we achieved beneficial biological effects with the magnetically assisted treatment translating the maintained light detection in *Bbs^−/−^* mice (KO). To our knowledge, this is the first demonstration of efficient magnetic drug targeting in the photoreceptors in vivo after topical administration. This non-invasive, needle-free technology expands the application of SMDs for the treatment of a vast spectrum of retinal degenerations and other ocular diseases.

## 1. Introduction

The non-invasive, targeted delivery of therapeutic agents to the retina is utterly challenging. Access to the retina imposes either the use of systemic or invasive administration routes [1]. The first is often inefficient because retinal targeting is minimal, and can be associated with substantial off-target effects [2]. The second, which is mainly accomplished via intravitreal and subretinal injections, allows the local delivery of therapeutic agents but repetitive injections are needed to maintain the drug concentration in the retina. Although millions of patients are treated with this invasive and unpleasant method every year, intraocular injection involves rare but serious risks for the patient’s sight [3,4,5], such as retinal detachment, vitreal haemorrhage, endophthalmitis, cataracts and others. Owing to these reasons, the treatment of pediatric retinopathy patients with repeated intraocular injections is considered ethically unacceptable.

Bardet-Biedl syndrome (BBS) is a retinopathy and degenerative emblematic ciliopathy which is characterized by autosomal recessive inheritance (it is associated with 24 causative genes) [6]. It is clinically characterized by the association of features ranging from retinitis pigmentosa, polydactyly, renal dysfunction and obesity. The BBS-mediated primary loss of rod photoreceptors is followed by the later demise of cone photoreceptors [7]. Electroretinography is the clinical examination of choice to spot the syndrome because it might show early changes within the first two years of life, even though significant changes are rarely visible before the age of five [8]. Most BBS patients become legally blind by their second or third decade, and presently, no treatment is available for BBS-induced retinal dystrophy besides classical palliative approaches such as learning the Braille alphabet for young patients [9].

Previously, we and others [10,11] demonstrated using mouse models that BBS-associated retinal degeneration was related to the pro-apoptotic activation of the unfolded protein response (UPR) associated with the gradual flattening of elettroretinogram (ERG) recordings, the thinning of the outer nuclear layer (ONL), and endoplasmic reticulum (ER) swelling. Different Bbs KO mice models have been shown to share similar UPR-related responses [11]. Based on these findings, we successfully repositioned two historical drugs: namely valproic acid (VPA) and guanabenz (GBZ), an anti-epileptic and anti-hypertensive drug, respectively. VPA enhances the expression of the HSP70 chaperone that binds to immunoglobulin protein (Bip) and results in an increase of the protein-folding capacity within the ER, whereas GBZ inhibits the GADD34 phosphatase and maintains elevated levels of the phosphorylated Eukaryotic Initiation Factor 2 (p-eIF2α) to block CAP-dependent translation [12]. The resulting synergistic effect of these two drugs efficiently reduced protein overload in the ER by simultaneously reducing de novo protein synthesis and increasing the protein folding capacity. 

With our earlier proof-of-concept showing significant efficacy to maintain vision capabilities in vivo, our next challenge was to develop a non-invasive method of drug delivery to the posterior part of the eye. The method should be compatible with the pediatric patient population, and would hence render the GBZ–VPA therapeutic combination a plausible treatment for clinical use for BBS.

Magnetic drug delivery, which consists in the combined use of magnetic nanoparticles (MNPs) to carry drugs and a magnetic field to guide the MNPs towards a specific target in the body, has been used for a long time in research and clinics for a variety of applications. These include cancer therapy [13], the intracellular delivery of nucleic acid (Magnetofection™) [14], and the delivery of stem cells [15], and have been used in combination with clinical MRI scanners for the targeting of MNPs to deeper tissues in the body [16]. Colloidal iron-oxide based MNPs are used in the clinics for the treatment of anemia [17], for the treatment of cancer through hyperthermia [18], and as diagnostic tool for contrast enhancement in magnetic resonance imaging (MRI) [18].

Magnetic drug delivery can improve the drug permeation across different ocular tissues without damage in ex vivo settings [19,20]. In vivo, magnetic targeting and MNPs have been applied to target the retina [21,22] and other ocular structures in the anterior segment of the eye [23]. We know from these experiments that MNPs can be concentrated in the retina from the blood circulation with the help of a magnet placed in contact with the ocular surface. MNPs and magnetic targeting have also been employed to guide intravitreally injected, magnetized stem cells to the retina using a magnetic device placed between the sclera and the orbital space [15]. Some magnetic devices for the treatment of ocular diseases using MNPs have been patented [24,25].

Herein, we report the development and successful in vivo use of the magnetically assisted delivery of the GBZ–VPA combination to the mouse photoreceptors after topical administration. VPA and GBZ were bound to the MNPs’ surface through electrostatic interactions. We started by synthesizing and screening a library of iron-oxide-based MNPs for the loading of VPA and GBZ through electrostatic interactions. Two MNPs, one for VPA and one for GBZ loading, were selected, and their physico-chemical properties were characterized. Then, the two drug-loaded MNPs were formulated as MNP-based eye drops, and were tested for toxicity in two different retinal cell lines. The GBZ- and VPA-loaded MNPs were then applied topically onto the ocular surface of wild type (WT) and *Bbs^−/−^* mice models, and were magnetically delivered to the retina for the treatment of the photoreceptors.

## 2. Materials and Methods

### 2.1. Reagents

Guanabenz acetate (GBZ) (Catalog # 0889, Tocris Bioscience, Ellisville, MO, USA) and valproic acid (VPA) (Catalog # 4543, Sigma-Aldrich, Saint-Quentin Fallavier, France) were purchased to make stock solutions of 1.25 mM GBZ and 100 mM VPA. They were then diluted with saline solution at pH = 6 to reach a final concentration of 2.5 μM GBZ and 0.2 mM VPA prior to their injection.

Iron(II) chloride tetrahydrate and ammonium hydroxide, 28–30%, were purchased from Acros Organics (Geel, Belgium); Iron(III) chloride hexahydrate tetraethyl orthosilicate (TEOS), 3-(tryhydroxysilylpropyl) methyl-phosphonate monosodium salt (42%) in water (THPMP), Zonyl^®^ FSA, hydrocortisone 21-hemisuccinate, putrescine, progesterone, β-mercaptoethanol, polybrene, polysine and polyethyleneimine were purchased from Sigma-Aldrich (St Quentin-Fallavier, France). All of the other chemicals and solvents were purchased from VWR (Les Ullis, France). All of the purchased reagents were used without further purification.

### 2.2. Synthesis of the MNPs

Iron oxide nanoparticles were synthesized by the classical co-precipitation of iron salts following the Massart method [26], with a few modifications as reported elsewhere [27]. Briefly, ferrous and ferric ions in solution (1:2 stoichiometry) were precipitated in an oxygen-free atmosphere by the addition of ammonium hydroxide. The mixture was heated up to 80 °C and maintained at this temperature for 30 min while stirring vigorously.

For the synthesis of the anionic MNP (NP01-NP03), TEOS or a combination of TEOS and THPMP or citric acid were added in one shot, and the mixture was stirred for 2 h. Thus, surface coating resulted from the in situ co-condensation of two silanes, yielding a silicon oxide layer presenting negative surface phosphonate groups. Then, the reaction result was purified by magnetic separation and washed thoroughly with MilliQ water until pH 7 was reached. The suspension was sonicated using a Branson Sonifier 250 (Branson Ultrasonics BV, Eemnes, The Netherlands), and was dialyzed against MilliQ water using a dialysis membrane with a molecular weight cut-off of 50 kDa (Spectrum Labs, Breda, The Netherlands), followed by another cycle of ultrasonication to yield dark-brown ferrofluids.

For the synthesis of the cationic MNPs (NP04-NP08), the capping of the crystals was achieved either by the addition of Zonyl^®^ FSA followed by the addition of a solution of cationic polymer such as polybrene, poly-L-lysine, polyethylenimine or proprietary ionizable amphiphilic block co-polymers, or using a mixture of TEOS and organosilanes carrying amine groups (i.e., APTES). Then, the crude reaction was purified by magnetic separation and washed thoroughly with MilliQ water until pH 7 was reached. The suspension was sonicated for 10 min and dialyzed against MilliQ water using a dialysis membrane with a molecular weight cut-off of 50 kDa (Spectrum Labs, Breda, The Netherlands) followed by another cycle of ultrasonication to yield a dark-brown ferrofluid. The MNPs’ iron content was quantified by colorimetric assay, as described elsewhere [27].

### 2.3. Analysis of the MNPs’ Colloidal Properties

The colloidal properties of the MNP ferrofluids were characterized by dynamic light scattering (DLS) and laser doppler electrophoresis using a Zetasizer Nano ZS (Malvern, Cambridge, UK).

A 50 µL volume of the MNP (1 mg Fe/mL) formulation was diluted to 1 mL with MilliQ water in disposable cuvettes. All of the samples were equilibrated at 25 °C before the analysis. DLS measurements were run first, followed by Zeta Potential experiments that were carried out using Malvern DipCell.

The hydrodynamic size of NP01, NP04 and their mixtures in different vehicles was measured in DPBS without calcium and magnesium, 0.9% wt/v sodium chloride and 5% wt/v glucose. For the analysis of the single MNPs, a volume of 25 µL of MNPs was suspended and diluted in a prefiltered (0.2 µm mesh size) drug loading solution to 50 µL. The resulting dispersion of MNPs was then diluted to 1 mL in a sterile cuvette using MilliQ water. For the MNP mixture, a volume of 15 µL NP01 in GBZ was gently added to 15 µL NP04 in VPA in a sterile cuvette. Then, the mix of MNPs was gently diluted with 30 µL of vehicle to reach the final volume of 60 µL. Next, the resulting MNP dispersion was diluted to 1 mL in a sterile cuvette using MilliQ water. Great care was taken to carry out the whole procedure under sterile conditions. Then, the evolution of the hydrodynamic diameter was measured as a function of time. The samples were monitored for at least 4 days. Before each measurement, the colloids were gently stirred by hand.

The study of the evolution of the hydrodynamic sizes of NP01, NP04 and their mixtures at different MNP concentrations was carried out as described above, except that different concentrations of the MNPs (200, 20 and 10 µg Fe/mL) were prepared by diluting the MNPs in the respective drug loading solutions. The diluted MNPs (25 µL) were gently diluted further with the vehicle (25 µL) to reach a final volume of 50 µL. These samples were diluted to 1 mL with MilliQ water. Great care was taken to carry out the whole procedure under sterile conditions. Then, the evolution of the hydrodynamic diameter was measured as a function of the MNPs’ concentration.

### 2.4. Determination of the Magnetic Core Size and Shape by Trasmission Electron Microscopy

The MNPs’ morphology, size and distribution were characterized by TEM using a JEOL 2100 (JEOL, Montpellier, France). The samples were sonicated for 30 min in an ultrasound bath (Bioblock Scientific, ThermoFisher Scientific, Strasbourg, France). Then, 1–2 µL of the single MNP was placed on holey carbon film-covered copper grids and left to dry before the exposition to the electron beam accelerated at 200 kV. The average MNP’s size and cumulative size distribution were obtained by measuring 100 single objects.

### 2.5. Determination of the Magnetic Core’s Crystalline Structure by X-ray Driffraction

The magnetic core’s crystallinity and chemical composition were analysed by powder X-ray diffraction (XRD) using a Bruker D8 Focus diffractometer (Bruker, Palaiseau, France). The instrument was equipped with 10 LynxEye detectors, and a Co source (Kά_1_ = 1.7887 Ȧ) working at 35 kV and 45 mA was used. An X-Y-Z stage configuration and goniometer PW3050 (Theta/Theta) were employed. The measurements were recorded in the 15–115 θ–2θ range, with a step size of 0.02 and 1.2 s/step acquisition. Equivalent amounts of freeze-dried samples were loaded onto a plastic slit and rotated at 15 rpm at 25 °C. As the reference material, a single crystal of quartz was ground, and its diffraction pattern was recorded over the same θ–2θ range. The fitting of the diffractogram was carried out using X’Pert Higscore software (Malvern Panalitycal, Malvern, UK).

### 2.6. Drug Loading Procedure and Quantification

Two stock solutions, GBZ and VPA, were prepared in MilliQ water and filtered through a 0.2 µm filter. Then, the stock solutions were diluted to prepare the loading solutions, one for GBZ (15 µM) and one for VPA (12 mM). Then, the MNPs were diluted to 1 mg Fe/mL with the respective loading solutions, vortexed for 5 min, and incubated at room temperature overnight.

To quantify the drug adsorbed on the surface of the MNPs, a high-performance liquid chromatography Infinity 1260 (HPLC, Agilent, Montpellier, France) was used. It consisted of the following elements: a quaternary pump VL Infinity with an integrated degaser unit; a thermostated column compartment, 1260 infinity; a Zorbax Plus—C18 column (100 mm × 4.6 mm, 3.5 µm); a UV detector, 1260 Infinity (190–650 nm); and a manual injector, 1260 Infinity, with a 20 µL loop for the injections. The mobile phase was of a water/methanol mixture (*v*/*v*, 70:30). Both solvents contained 1% formic acid (FA), and were degassed by ultrasonication for 20 min and further filtered through a 0.2 µm filter. The column was maintained at 30 °C for both analytes, and the drugs were detected at UV wavelengths (270 nm for GBZ and 210 nm for VPA). The retention times and peak areas were recorded and calculated using OpenLAB CDS LC ChemStation software (Agilent, Santa Clara, CA, USA). For each sample, triplicates were performed. Detailed information on the elution protocols and the analytes’ detection for GBZ and VPA are reported in Appendix A.

Serial dilutions were prepared from the stock solutions (see above) and injected into the HPLC apparatus to establish calibration curves for the drugs (Appendix A). Thereafter, 500 µL of the formulations and 500 µL of the pure loading solutions (blanc) were centrifugated separately at 3000 rpm for 30 min in Amicon^®^ Ultra-4 centrifugal filters with a 100 KDa molecular weight cut-off (Millipore Merck, Molsheim, France). Subsequently, the filtrates were injected into HPLC, and the drug concentration was deduced using the calibration curve. Then, the quantity of the drug adsorbed on the MNPs’ surface was obtained by the subtraction of the drug concentration in the filtrate obtained from the MNPs’ base formulation (*Cf*) from the drug concentration in the blank (*Cb*). Finally, the loading efficiency (*LE*) of the loading process was calculated using the following equation:(1)LE=[(Cf/Cb)/Cb]∗100

### 2.7. Cell Culture

Two cell lines were used: 661 W, a photoreceptor precursor cell line (The University of Oklahoma, Health Sciences Center, Department of Cell Biology), and hTERT-RPE1 cells (ATCC^®^ CRL-400). The cells were cultured in T75 plates at 37 °C, 5% CO_2_. The culture media for the 661 W cells was DMEM (ThermoFisher Scientific, Munich, Germany) supplemented with hydrocortisone 21-hemisuccinate, progesterone, putresceine, β-mercaptoethanol, 10% fetal bovine serum (FBS) (ThemoFisher Scientific, Munich, Germany) and antibiotic–antimyotic (ThermoFisher Scientific, Munich, Germany). The culture medium for the RPE1 cells was DMEM:F12 (ThermoFisher Scientific, Munich, Germany) supplemented with 10% FBS and hygromycin B.

### 2.8. In Vitro Cell Viability Assay

The cell viability was studied using an OZ Blue Cell Viability kit (OZ Biosciences, France) following the manufacturer’s indication, with slight modifications. Briefly, one day prior to the assay, 661 W and RPE1 cells were trypsinized and seeded at 20,000 cells per well in a 96-well plate. The next day, both cell lines were exposed to increasing concentrations of the two MNPs (loaded and unloaded with drugs) and the MNPs mixture formulation in 5% glucose. The vehicle alone (5% glucose) was used as a control. After the addition of the MNPs into the wells, the plate was placed on a magnetic plate (OZ Biosciences, #MF14000) for 15 min at 37 °C to perform the magnetic targeting, as in Magnetofection™, as described elsewhere [27]. Then, the magnetic field was removed and the cells were incubated for 24 h. Thereafter, the cell viability was monitored using an OZ Blue Cell viability kit through fluorescence measurement (560 nm_Ex_/590 nm_Em_) with a cytofluor multi-well plate reader series 4000 (Perseptive Biosystems, Framingham, MA, USA). The experiments were repeated in triplicates. The results were expressed as the percentage compared to the control condition (cells treated only with vehicle, glucose 5%).

### 2.9. Mouse Generation and Husbandry

All of the experimental procedures were approved by the local ethical committee of Strasbourg University, with the authorization number #22448-2019101515488464. *Bbs1^M390R/M390R^, Bbs10^−/−^* and *Bbs12**^−/−^* mice, and their control *Bbs^+/+^* (wild type, WT) littermates, were generated as described previously [10,12,28]. The relevant mouse models were bred on a C57/B6N background, and were cross-bred with the C57/C6J strain to remove the strain-associated interfering Rd8 mutation which interferes with the retinal phenotype [29]. The mice were kept and bred in humidity- and temperature-controlled rooms on a 12-h light/dark cycle, on normal chow and water *ad libitum*. These three mouse models share the same UPR mechanism behind the apoptosis of the photoreceptors, and similar disease progression [11,12]. An extensive table with the mice and their genetic backgrounds in the experiments is found in Appendix A.

### 2.10. Preparation of the Eye Drops for the Topical Application of the MNPs

For the in vivo experiments, the MNP eye drops were prepared as a mix of NP01 and NP04. Each eye drop had a volume of 20 µL, and they were prepared as follows: First, 5 µL NP01 (1 mg Fe/mL) loaded with GBZ was gently added to 5 µL NP04 (1 mg Fe/mL) loaded with VPA. The mix was further diluted with 10 µL 10% glucose, followed by gentle pipetting. The eye drops were always prepared fresh before the instillation.

For the preparation of the unloaded MNP eye drops, we used the same proportions: 5 µL unloaded NP01 (40 µg Fe/mL) + 5 µL unloaded NP04 (40 µg Fe/mL) and 10 µL 10% glucose vehicle. For the preparation of the eye drops with only GBZ or VPA, the protocol was the same, but only one type of loaded MNP was used, and the other MNP was without a drug payload.

For the preparation of the eye drops with different concentrations of MNPs, both NP01 and NP04 were diluted separately before mixing using the same loading solutions used during the loading procedure (15 µM GBZ for NP01; 12 mM VPA for NP04). Once the MNPs were diluted to 40 µg Fe/mL (1:25 dilution) and 20 µg Fe/mL (1:50 dilution) they were mixed following the same protocol mentioned above.

The mix of GBZ and VPA loading solutions, 15 µM and 12 mM respectively, without any MNPs were also used as a control (GBZ+VPA). The proportions of GBZ and VPA in the eye drops were the same as above.

### 2.11. Topical Application of the MNP-Based Eye Drops under the Influence of a Magnetic Field

The topical application of MNP-based eye drops was performed in 14-days old mice under anesthesia using Domitor^®^ (Medetomidine, 6.5 µg/g body weight) and Ketamine (665 µg/g body weight). The list with the different treatment groups is found in Appendix A. Topical application procedure consisted of the following steps: (1) 10 µL eye drop was applied on the right eye, and the magnet was placed behind the head of the mouse; (2) a second 10 µL eye drop was gently applied to the same eye 5 min later; (3) the untreated left eye (control eye) was covered with Ocrygel^®^ to avoid corneal drying; (4) the magnetic targeting was continued for 25 min after the second eye drop’s instillation (for a total of 30 min). Thereafter, the magnet was removed, and the leftover of the eye drop was carefully wiped. Ocrygel^®^ was applied to both eyes until the mice were fully awake.

### 2.12. Electroretinography

Electroretinograms (ERGs) were recorded two weeks after the topical treatment with MNP-based eye drops using an HMsERG system (Ocuscience^®^, Kansas City, MO, USA). The mice were dark-adapted overnight and then anesthetized by the intraperitoneal injection of Domitor (7.6 µg/g body weight) and ketamine (760 µg/g body weight). The pupils of the animals were dilated with 0.3% atropine eye drops before the ERG recordings. The experiments were carried out in dim red light (Catalog # R125IRR, Philips, Suresnes, France). A standard ERG procedure was used according to the manufacturer’s protocol (Ocuscience^®^, Kansas City, MO, USA). Briefly, the protocol consisted of a recording using a dark-adapted ERG (scotopic ERG) after photonic stimuli with intensities ranging from 0.1 to 25 cd.s/m^2^. The ERG results were amplified and captured digitally using an ERG View system 4.3 (Xenotec, Ocuscience^®^, Kansas City, MO, USA). The a- and b-waves of the scotopic responses were recorded at post-natal day 28, two weeks (14 days) after the treatment with MNPs.

For the study of the MNPs’ safety in wild-type mice (*Bbs^+/+^)*, a total number of 23 *Bbs^+/+^* animals at post-natal day 14 were divided into 4 groups (three treated groups of 6 mice each, and one untreated group of 5 mice). The groups were organized as follows: untreated wild-type mice, mice treated with non-diluted MNP eye drops (1 mg Fe/mL), and mice treated with diluted MNP eye drops at a concentration of 40 µg Fe/mL) and 20 µg Fe/mL. In the treated groups, only the treated eye was analyzed, while both eyes of the untreated groups were analyzed with ERG. Thus, 6 independent ERG records were recorded for each treated group, and 10 ERG records were recorded for the untreated group.

After the safety study in the *Bbs^+/+^* mice, the MNP formulation with 40 µg Fe/mL was chosen to assess the treatment effects in the *Bbs^−/−^* mice.

For the study of the GBZ and VPA delivery to the photoreceptors of *Bbs^−/−^* mice, a total number of 35 animals were divided into 6 groups: five treated groups each composed of 6 animals, plus a control group composed of 5 animals. For the treated groups, only the treated eye was analyzed in the ERG, while for the untreated groups, both eyes were used for the ERG analysis. Thus, a total of *n* = 6 eyes were analysed for the treated groups, and *n* = 10 eyes were analysed for the untreated groups.

The groups were organized as follows: *Bbs^−/−^* which were not treated (untreated); *Bbs^−/−^* treated with MNP mixed with GBZ and VPA (40 µg Fe/mL MNPs); *Bbs^−/−^* treated with an NP01+NP04 mix without the drug (unloaded 40 µg Fe/mL MNPs); *Bbs^−/−^* treated with GBZ and VPA, but without MNPs; and *Bbs^−/−^* treated with the MNP mix (40 µg Fe/mL MNPs), but loaded with either GBZ or VPA.

After ERG, half of the eyes were dissected from the animals and prepared for TEM imaging.

### 2.13. Histological Analysis by Light Microscopy and TEM

The dissected eyes were fixed by immersion in 2.5% glutaraldehyde and 2.5% paraformaldehyde in a cacodylate buffer (0.1 M, pH 7.4), post-fixed in 1% osmium tetroxide in 0.1 M cacodylate buffer for 1 h at 4 °C, and dehydrated through graded alcohol (50, 70, 90, 100%) and propylene oxide for 30 min each. The samples were embedded in Epon™ 812 (Sigma-Aldrich, Saint-Louis, MO, USA).

For the TEM imaging, ultra-thin sections were cut at 70 nm, contrasted with uranyl acetate and lead citrate, and examined at 70 kv with a Morgagni 268D electron microscope. For our experiments, we studied the eyes of three different mice per group (untreated WT, untreated KO mice, KO mice treated with 1 mg Fe/mL and 40 µg Fe/mL MNPs mixed in 5% glucose). A total of three different images were assessed for each of the mice. The images were captured digitally using a Mega View III camera (Soft Imaging System).

### 2.14. Ocular Biodistribution of the MNPs

The biodistribution of the MNPs was studied with magnetic resonance imaging (MRI). A total of 9 two-month-old *Bbs* WT mice were divided into three groups, each composed of 3 animals. One group did not receive any treatment, the second group was treated as described before with the use of magnetic targeting, and the third group was treated like the second, but without the magnet. Due to the sensitivity limits of MRI imaging, a lower volume with a higher concentration of MNPs was used. The preliminary experiments with the standard MNP-based eye drops (see above) resulted in too low a signal/noise ratio and a lack of reliable imaging. Therefore, the eye drops used for the MRI experiments consisted of a single application of 0.5 µL eye drops containing a mix of both MNPs at 6 mg Fe/mL for each (0.25 µL of NP01+ 0.25 µL of NP04). After 45 min of magnetic targeting, the distribution of the MNPs was studied using MRI (7T Bruker Pharmascan, Ettlingen, Germany). The MRI images were acquired right after the treatment without extra anaesthesia. Three-dimensional T_2_-wighted data (RARE-sequence, repetition time 1.2 s, effective echo time 29.6 ms, a train of 16 echies/excitation with 3.7 ms between the echoes, field of view 6.4 × 6.4 × 4.8 mm, matrix 64 × 64 × 48 yielding a 100 µm spatial resolution) were collected separately from each eye using a volume transmit coil and a 10 mm surface receiving coil placed on top of the eye. Each eye was imaged separately. The animal groups are shown in Appendix A.

### 2.15. Statistical Analysis

Student’s *t*-tests were applied to all of the datasets with two tails (two samples). Non-parametric tests (Mann-Whitney tests) were applied to all of the datasets with two tails (two samples) for the in vivo experiments. All of the data in the bar charts show the mean ± SEM. *p* < 0.05 (*) is considered to be a significant difference.

## 3. Results

### 3.1. MNP Screening for Optimal Drug Loading

The loading of the MNPs was optimized based on the electrostatic binding of the negatively-charged VPA and the positively-charged GBZ to the MNPs. We investigated the impact of various MNP parameters: the type of coating, electrostatic charge, hydrodynamic size, and polydispersity index (PDI) (Table 1). All of the MNPs had a hydrodynamic size in the nanoparticle range, and the surface charge was controlled with the coating (Table 1). A total of eight MNPs, three negatively charged MNPs for the loading of GBZ (NP01, NP02 and NP03) and five positively charged MNPs for the loading of VPA (NP04, NP05, NP06, NP07 and NP08), were tested for the loading of GBZ and VPA, respectively.

NP03 and NP01 were the smallest MNPs (125 ± 53.9 nm and 142 ± 64.7 nm, respectively) (Appendix A). All of the other MNPs had hydrodynamic sizes between 150 nm and 250 nm. In terms of their zeta potential, the MNPs varied from −44 mV (anionic) to +38 mV (cationic) (Table 1).

The loading experiments (see Table 1 and Appendix A) showed significant drug binding with almost all of the MNPs (Table 1, Appendix A). NP02 was an exception, because GBZ’s addition resulted in the rapid sedimentation of the magnetic colloid (thus, NP02 was excluded). Maximal drug loading was used as the criterion in the selection of the MNPs. Among the anionic MNPs (for GBZ loading), NP01 was selected (with a loading efficiency of 31.5 ± 15.7% or 3.99 ± 1.99 µM). For the VPA loading, cationic MNP NP04 was selected (with a loading efficiency of 11.8 ± 3.1% or 1.18 ± 0.31 mM).

### 3.2. Physicochemical Characterization of the Selected MNPs

Figure 1A shows the principle of the electrostatic loading of GBZ and VPA onto the MNPs. The addition of electrolytes, like GBZ and VPA salts, might lead to the aggregation of colloids that are normally stabilized by repulsive electrostatic interactions [30]. Therefore, we studied the influence of the drug loading on the zeta potential of NP01 and NP04. Figure 1B,C shows the zeta potentials and the hydrodynamic sizes of the MNPs with and without the drugs (BBZ, VPA). The results demonstrate that GBZ and VPA loading did not affect the zeta potential of the MNPS, and it did not induce the aggregation of NP01 and NP04, respectively.

Additionally, we investigated whether improved drug loading could be achieved by increasing the GBZ and VPA concentrations on the fixed MNP levels (1 mg Fe/mL) (Appendix A). Higher drug concentrations induced the aggregation of both magnetic colloids; thus, this option was not considered any further.

Moreover, MNP-based therapeutics should avoid magnetically-induced particle aggregation. Nano-sized magnetic materials become magnetized in a magnetic field and lose the magnetized feature when the field is removed. Therefore, NP01 and NP04 were characterized in terms of their crystalline structure by X-ray diffraction (Appendix A), and in terms of both their size and morphology by transmission electron microscopy (TEM) (Appendix A). Both MNPs’ cores were composed of a magnetic iron oxide made of magnetite, the clinically used polymorphic form of iron oxide [17,18,31]. This is the form of iron oxide that yields the strongest magnetization in the presence of an external magnetic field [32]. Both NP01 and NP04 have spherical cores with average diameters of 8 nm and 10 nm, respectively (Appendix A). This information confirms the superparamagnetic nature of the NP01 and NP04 cores, as it is known that magnetite cores with sizes below 20 nm display superparamagnetism [32,33].

### 3.3. Formulation of the Drug-Loaded MNP-Based Eye Drops

The simultaneous administration of GBZ and VPA was shown to have synergistic preserving effects on the retinae of *Bbs-*deficient mice [12]. Accordingly, we need to formulate together two oppositely charged nanoparticles loaded with GBZ and VPA in a single eye drop without particle aggregation. First, we tested three vehicles: 5% glucose, PBS, and 0.9% sodium chloride. The stability of each particle separately (NP01, NP04) (Appendix A) and together (NP01+NP04) (Figure 2A,B) was studied in the three vehicles. Both the hydrodynamic size (Z-average) and the PDI increased in the presence of electrolyte-containing vehicles (sodium chloride, PBS), demonstrating particle aggregation (Figure 2A,B). In contrast, no aggregation over 4 days was seen in the presence of the 5% glucose vehicle (Figure 2A,B). This is an important finding which supports the use of a glucose-based vehicle that avoids the aggregation of two MNPs with opposite charges in the eye drops.

Controlling the size distribution of the MNP colloids is fundamental to obtain reproducible results for human trials. More details on the particle population can be obtained by comparing the size distribution plots as a function of time. In Figure 2C, the plots of the NP01+NP04 mix in the glucose vehicle show two peaks, revealing two distinct populations of MNPs. These plots showed that the relative intensity of the nanosized fraction decreased from 98.7% to 85.5% over four days, suggesting a slow tendency of MNP aggregation (Figure 2C,D). Importantly, at four days, the average hydrodynamic size of the nano-sized peak increased only by 13 nm, suggesting the stability of the nanometric MNP population. In contrast, sodium chloride and PBS vehicles lead to the massive aggregation of the MNPs (Appendix A). Based on these findings, we used freshly prepared MNP-based eye drops in 5% glucose in the ocular experiments.

Colloids stabilized by electrostatic interactions are known to aggreate if the equilibrium between the solid and liquid part of the system is altered. MNPs stabilized by electrostatic interactions are known to aggregate upon dilution [34,35], which is an eventuality we excluded for our MNP mixture in glucose by DLS measurements (Appendix A).

### 3.4. Biocompatibility of the MNP-Based Therapeutics In Vitro

Next, we investigated the possible toxicity of the MNPs in living cells. The biocompatibility of NP0-, NP04- (Appendix A) and MNP-based eye drops in the glucose vehicle (Figure 2E,F) was assessed in vitro in a murine photoreceptor cell line (661 W) and human retinal pigmented epithelial cells (RPE1). The cells were exposed to the MNP mix (0.1–10 µg Fe/mL) and magnetic targeting (30 min). The cell viability assessment showed that 661 W cells tolerated the formulation even at 10 µg Fe/mL (Figure 2E). The RPE1 cells were more sensitive than 661 W, but the MNP exposure decreased the cell viability significantly only at the concentration of 5 µg Fe/mL for both the loaded and unloaded MNP mixture (Figure 2F). However, the decrease in the cell viability observed for the 10 µg Fe/mL (higher MNPs concentration) concentration was less significant.

### 3.5. Upon Topical Eye Drop Administration, MNPs Are Successfully Transported to the Retina with a Magnetic Push

Next, we studied the ocular biodistribution of the MNPs using a clinical MRI scan. Figure 3A shows the experimental setup used for the ocular application of the MNPs, and the magnetic targeting. The MNP-based eye drop is instilled on the right eye, while the magnet is placed behind the head of the mouse to maximize the magnetic pull towards the retina. Figure 3B illustrates the position of the different planes used for acquisition of the MRI images respective to the mice.

An MNP-based (NP01+NP04) eye drop mix was instilled on the right eye (treated eye), while the left eye (control) was covered with Ocrygel. The magnetic targeting was carried out for 45 min by placing a 0.4-Tesla magnet behind the mouse’s head, as shown in the picture. In order to assess the ocular distribution of the MNPs, wild type C57/C6J mice were used to verify the cell penetrability in intact eyes. The MNP-based (NP01+NP04) eye drop mix was used at a 6 mg Fe/mL (see Methods). The overall ocular distribution of the MNPs was assessed using MRI at 7T. These experiments were carried on two-month-old mice because their eyes are sufficiently large enough to allow the optimal detection of the MNPs. Figure 3C shows MRI images of mice treated with a single dose of MNP-based eye drops, both with and without magnetic targeting. The MNPs are indicated by a red arrow, while the lens is indicated by a white arrow and the eyeball by a black arrow. Here, we wanted to assess whether the influence of the magnet was able to increase the penetration of the MNPs into the ocular tissues. As shown in the MRI images of the mice treated while under the influence of the magnet, the amount of MNPs inside the ocular tissues is higher than the amount of MNPs penetrating the ocular tissue without the help of the magnetic push.

Although the MRI showed the improved ocular penetration of the MNPs in vivo when magnetic targeting is used, this method does not resolve the issue of single MNPs and their exact localization in the tissues. Thus, we repeated the same experiment as before, but using 14-days postnatal WT C57/C6J mice to match the experimental conditions of our previous study [12]. Furthermore, we used more diluted MNPs (1 mg Fe/mL) because of the higher resolution of the TEM method compared to the MRI. Then, the intraocular distribution of the MNPs was determined by histological analysis using transmitted electron microscopy (TEM). Figure 4A shows representative TEM images acquired from animals which did not receive any treatment. In the animals that received the topical administration of the MNP-based treatment (NP01+NP04) followed by 30 min of magnetic targeting (Figure 4B), the iron-oxide cores of the MNPs were spotted (red arrows) in different retinal layers, including the photoreceptor ONL, OS and IS, thereby confirming the non-invasive MNP targeting in our cellular target. These results evidenced that magnetic targeting improved the MNPs’ distribution through multiple retinal layers.

### 3.6. Non-Invasive, Magnetically Assisted GBZ/VPA Delivery Improves Retinal Function in Bbs KO Mice

Our next step was to verify whether the delivery of VPA and GBZ with the MNPs (NP01+NP04) is effective in improving vision loss in the BBS. For that purpose, we carried out scotopic ERGs experiments in the Bbs mice.

Scotopic ERG records represent the summed activity of photoreceptors, mainly rods in mice, plus downstream retinal cells, providing an effective means to identify deficits in photoreceptor signaling [36]. The signalling in the outer retina (rod photoreceptors) is assessed based on the a-wave, while the b-wave reports the rod bipolar activity in the inner retina [36]. In order to assess possible adverse effects of the MNPS in the retina, the scotopic ERGs were performed in WT mice that received MNPs (NP01+NP04) at Fe concentrations of 20 µg|mL, 40 µg/mL and 1 mg|mL, while the GBZ and VPA levels were constant. The results obtained from these treated groups were compared to an untreated group (Appendix A). No significant change in the a-wave and b-wave amplitudes was observed, indicating that MNP-based eye drops and magnetic fields did not cause adverse effects on retinal function.

Subsequently, the delivery of GBZ/VPA MNPs (NP01+NP04) to the photoreceptors was assessed in *Bbs* KO mice (Figure 5). The test groups were as follows: (1) MNP mix (40 µg Fe/mL) treatment, (2) GBZ+VPA without MNPs, (3) unloaded MNP mix, (4) MNPs loaded with GBZ, (5) MNPs loaded with VPA, and (6) untreated Bbs KO mice. The choice of treating *Bbs* KO mice only with 40 µg Fe/mL was due to the difficulty of breeding these animals, which limited the number of the mice.

Figure 5A,B shows the average a-wave and b-wave amplitudes from the scotopic ERG records at a low luminance intensity (0.3 cd.s/m^2^). The average amplitudes of the a-wave and b-wave in the treated KO group (40 µg Fe/mL MNPs) were 20 µV and 63 µV, respectively, which were significantly higher levels than in the untreated group. Figure 5C,D shows the average a-wave and b-wave amplitudes in the control groups and the MNP-group (40 µg Fe/mL) at an equal flash intensity. GBZ+VPA without MNPs, MNPs without GBZ+VPA, and MNPs with one drug (GBZ or VPA) augmented neither the a-wave nor b-wave amplitudes. Thus, the improvement of the ERG waves was due to the magnetically assisted delivery of GBZ and VPA to the photoreceptors of the *Bbs^−/−^* mice. Figure 5E,F shows the average a-wave and b-wave amplitudes at all of the flash intensities in the untreated group, and after the application of MNPs (40 µg Fe/mL) with GBZ and VPA. The significant amelioration of the photoreceptors’ functionality with the MNPs was observed throughout the range of flash intensities. Appendix A shows typical scotopic ERG recordings obtained from *Bbs* WT mice and *Bbs^−/−^* mice (untreated and treated with 40 µg Fe/mL MNPs).

Finally, we wanted to investigate whether the MNP-mediated delivery of GBZ and VPA improved the ER dilatation in the inner segment (IS) of the photoreceptors, a well-established parameter related to BBS-induced protein overload inside the IS that results from the defects in ciliary transport. The TEM analyses in Figure 5G show the reduced ER in the IS of the photoreceptor (contoured in red) of the treated eyes.

## 4. Discussion

Non-invasive retinal drug delivery represents a major challenge. In direct contact with the eye, topical administration is the easiest way to apply drugs, but it is inefficient in retinal drug delivery, as only a negligible fraction of the applied dose (<0.01%) reaches the posterior segment of the eye [1]. Herein, we present an innovative approach for non-invasive retinal drug delivery. The approach is based on the magnetic pull of supraparamagnetic MNPs towards the posterior part of the eye. We presented a proof-of-concept for this new non-invasive delivery approach for retinal drug delivery.

Previously, preclinical attempts at retinal drug delivery by topical administration have been reported. They were based on several daily applications over prolonged periods [37,38]. Other groups used transscleral iontophoresis [39] to improve the intraocular delivery of topically administered drugs. This method induces physical alterations and potential damage to ocular barriers, and requires the use of sophisticated machines and trained technical staff. In contrast, the use of magnetic force does not require complex machinery or trained personnel. Furthermore, the presence of small liposomes was shown in the mouse retina, but not quantitated, after topical application [40]. Despite some promising results, the quantities of topically applied drugs and delivery systems in the retina are probably very low. This quantification requires the use of highly sophisticated methods, yet it was not the target of this early proof of concept study.

The use of MNPs in ophthalmology is rather recent and promising [14]. For example, Zhan et al. [41] showed that magnetic force facilitated the permeation of MNPs through porcine sclera ex vivo. Others [21,22] showed that intravenously injected MNPs accumulated to the mouse retina due to in the presence of a magnet right in front of the eye. Such an approach might be useful for the retinal targeting of drugs and even cells.

The results presented herein demonstrate the localization of MNPs in the retina, and the beneficial pharmacological actions of the loaded drugs (GBZ, VPA) on ERG readings in the BBS-deprived mice.

Despite these positive preclinical results, several problems must be overcome prior to human trials. For example, the exact migration route of the topical MNPs to the retina and the quantity of the drugs and MNPs in the retina are not known. Based on the published literature, MNPs have been shown to cross thicker explants (e.g., 500 µm thick porcine sclera) only in the presence of an applied magnetic force [19,41]. It is therefore possible that our MNPs would cross the mouse sclera (with an average thickness of 60 µm) to some extent. Further studies are needed to characterize the migration route of the MNPs, for example, by using fluorescent MNPs, the histological detection of MNPs [42] or optical coherence tomography.

The migration of the MNPs and the delivery of the cargo molecules are related to each other. We applied topically a single dose of 2 × 10^−8^ g and 8.65 × 10^−6^ g of GBZ and VPA, respectively, with the MNPs to the mouse eyes. However, we were able to improve the ERG parameters and state of the ER in the retina two weeks post-administration if both drugs were loaded to the MNPs, but not in other cases. We therefore conclude that a pharmacologically relevant amount of the VPA and GBZ is delivered to the retina. To fully characterize the exact amount transported to the retina by the MNPs, sensitive analyses (e.g., quantitative mass spectrometry and the liquid scintillation counting of radio-labeled drugs) is required.

Indeed, we attempted to determine the drug release from the MNPs (data not shown), but these experiments were not conclusive due to some technical limitations. The dilution of the drug in the release medium over the study decreased the drug concentraion below the limit of the dection of our HPLC. Thus, we lyophilized the fractions of release and rehydrate them with 1/10 of the original volume. This led to problems of solubility in the drugs, especially for GBZ, which has a rather poor solubility in water. Finally, the extraction of the drug from the release medium using organic solvents did not work, probably for the same reason as above. Although the drug-release properties of the MNPs were not studied further, it has to be stressed that the GBZ and VPA loading on the MNPs was mediated only by electrostatic interactions (Figure 1A). As was observed from other groups using similar drug delivery systems [43,44], a classic burst drug release profile must be expected from our method, leading to complete drug release in few minutes.

It is important to clarify here that MNPs do not improve the delivery of drugs because they physically pull the drugs into the tissues/cells under the effect of a magnetic field. MNPs are used to overcome diffusion-related barriers. In combination with magnetic targeting, MNPs are allowed to sediment and focus a large part of the therapeutic dose on the target within few minutes (rapid kinetics) [14]. This, in turn, boosts the tissue/cellular drug uptake, as was recently demonstrated in retinal explants where large siRNA was delivered in cells distributed across all of the retinal layers [45,46].

We also characterized the safety of our approach. The data on the photoreceptor functionality (ERG) post-treatment (Figure 4, Appendix A) did not show negative effects, suggesting that the MNPs did not impair retinal functions, a finding in agreement with the previously published data [42,47,48]. Previous long-term ocular toxicity studies in rats with nano- and microparticles suggest a good tolerance in ocular cells and tissues [15,42,47,48,49]. For example, nanoparticles did not show any toxicity after intravitreal and intracameral injections (i.e., no change in intraocular pressure, no activation of astrocytes or retinal glia and microglia) after 5 months [42,48]. The MNPs did not show any problems related to photoreceptor function, aqueous drainage or iron deposition [42].

In the future, the development of single-MNP formulations for sustained drug release is an option to be considered. For example, the coating of the magnetic nanocrystals with drug-loaded mesoporous silica shell is an option [50,51,52]. Although a single administration of MNP-based eye drops was shown to be beneficial in *Bbs* KO mice, the effect of repetitive aministration remains to be characterized, and then coupled with more in-deep studies related to MNPs biodistribution and drug-release profiles. Furthermore, it is important to test our drug delivery system in larger animals, such as rabbits. It must be noted that retinal drug delivery after topical application in mice is easier than in rabbit or human eyes. This is due to the thinner membrane barriers and shorter distances in the mouse eye compared to larger eyes. Lastly, the magnetic field parameters should be optimized to maximize the retinal delivery in rabbit and human eyes. The magnetic guidance of the MNPs can be foreseen as a pull system as in the case of the present study and others [15,21,22], but a push system has also been proposed [53].

## 5. Conclusions

Presented herein is the proof-of-concept study on magnetically assisted and topically delivered guanabenz and valproic acid in mouse eyes. We conclude that magnetic nanoparticles provide non-invasive drug delivery to the mouse photoreceptors to such an extent that a therapeutic effect was seen in the animal model of emblematic BBS ciliopathy (*Bbs* knockout mice). This approach did not cause any toxic effects in the eyes. The same approach may be applicable for the ocular delivery of other drugs. This approach to non-invasive ocular drug delivery warrants further studies in larger animal eyes prior to possible human studies.

## Figures and Tables

**Figure 1 pharmaceutics-13-01650-f001:**
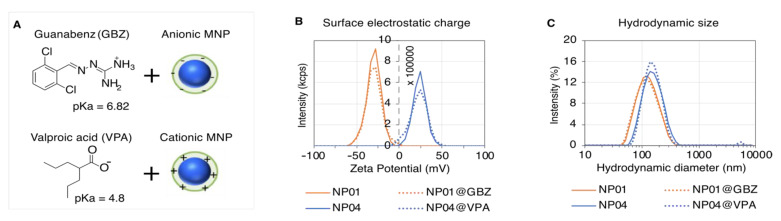
GBZ and VPA loading on the MNPs. (**A**) Illustration of the drug loading strategy exploiting the attractive electrostatic interaction between the ionized groups of GBA and VPA, and the charges of the opposite sign localized at the surface of anionic or cationic MNPs, respectively. Plots of the zeta potential (**B**) and hydrodynamic size (**C**) distributions of NP01 (orange plot), NP01 loaded with GBZ (orange dotted plot), NP04 (blue plot) and NP04 loaded with VPA (blue dotted plot). The zeta potentials of NP01 and NP01 loaded with GBZ (NP01@GBZ) are −29.8 ± 9.5 mV and −29 ± 10.2, respectively, whereas the zeta potentials of NP04 and NP04 loaded with VPA (NP04@VPA) are 3.0 ± 8.03 mV and 31 ± 11.3 mV respectively. The average sizes of NP01 and NP01@GBZ are 141.8 ± 64.7 nm and 146.1 ± 7.0 nm, respectively. The average sizes of NP04 and NP04@VPA are 169.7 ± 63.8 nm and 171.1 ± 61.3 nm, respectively.

**Figure 2 pharmaceutics-13-01650-f002:**
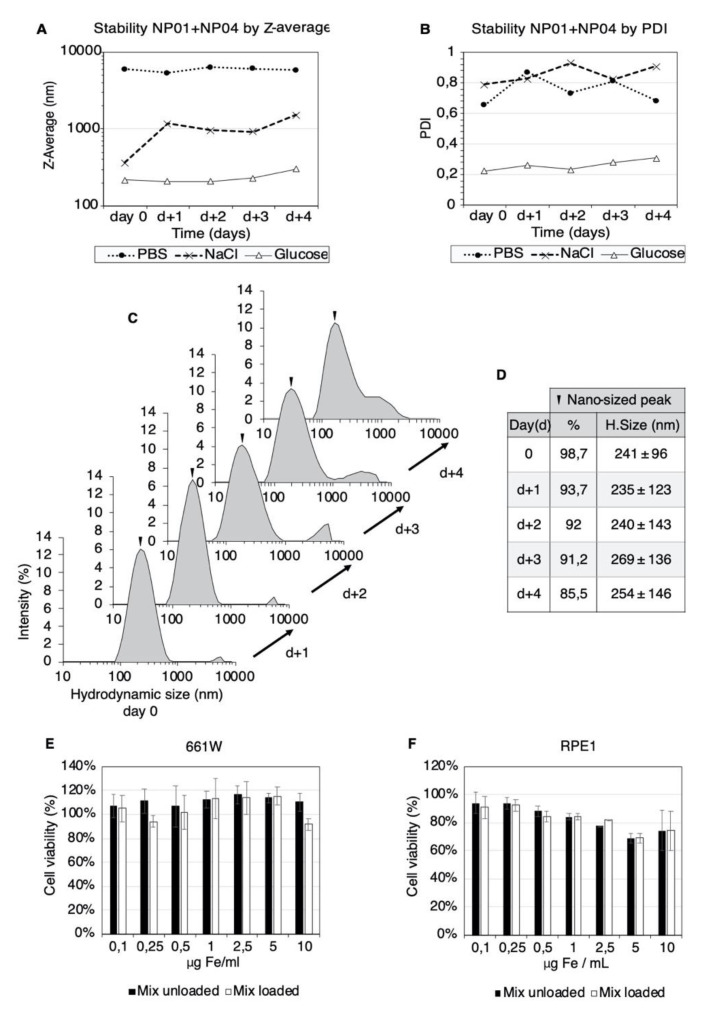
Formulation and biocompatibility of the MNP-based eye drops. Plots of the intensity-weighted mean hydrodynamic size (Z-average) (**A**) and poly dispersity index (PDI) (**B**) of the NP01+NP04 mixtures at different times. Three vehicles were used: 5% glucose (black line), PBS (dotted line) and 0.9% sodium chloride (dashed line). The evolution of the hydrodynamic size of the NP01+NP04 mixture in 5% glucose over four days (**C**). The hydrodynamic size and contribution of the nanosized particle population (**D**). The viability of 661 W (**E**) and RPE1 (**F**) cell lines after exposure to the NP01+NP04 mixture in 5% glucose and 30 min of magnetic targeting. The measurements were carried out after 24 h of incubation. The mean ± SD (*n* = 3) from three independent experiments are shown. The results are expressed as a %, using the cells treated with vehicle alone (glucose 5%) as control.

**Figure 3 pharmaceutics-13-01650-f003:**
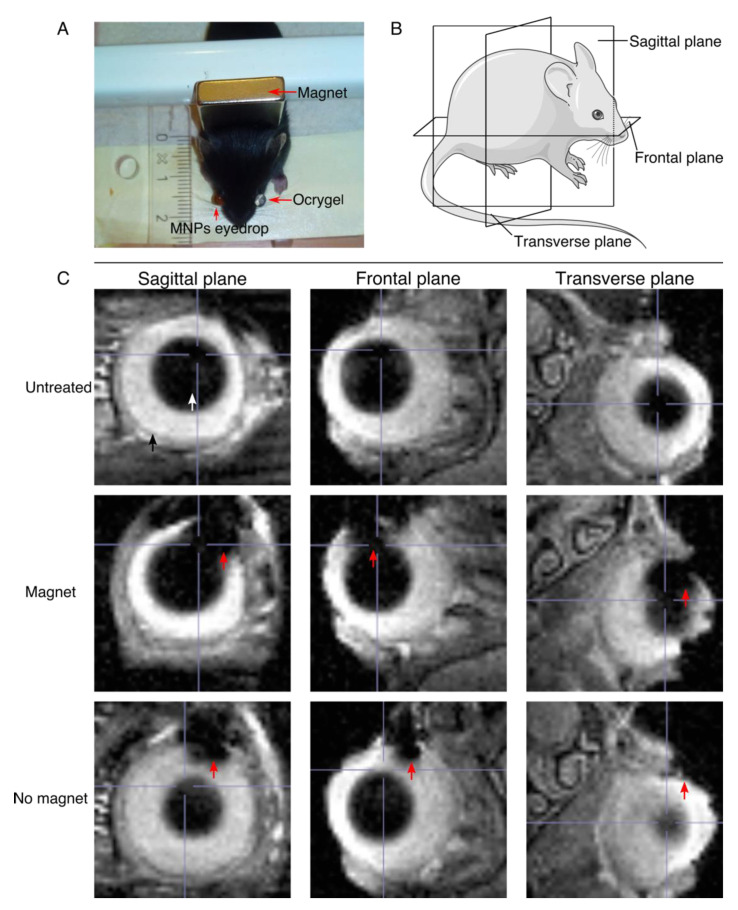
MRI image acquisition and MRI images showing the penetration of the MNPs into the ocular tissues. (**A**) Illustration of the experimental setup for the mouse experiment. (**B**) Illustration of the different planes used in the MRI image acquisition. (**C**) MRI images showing the penetration of the MNPs into the ocular tissues. The three different planes of acquisition are shown for the untreated mice and mice treated with and without the influence of the magnetic field. The red arrow points at the MNPs, the white arrow points at the lens, and the black arrow points at the eyeball.

**Figure 4 pharmaceutics-13-01650-f004:**
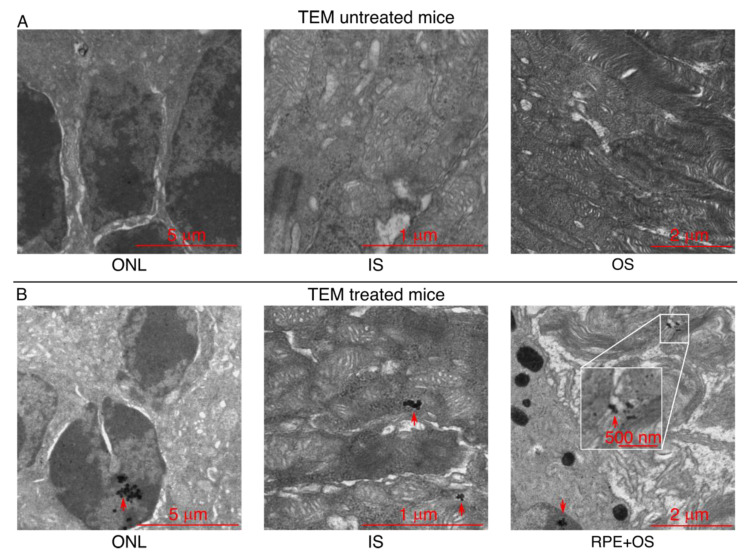
MNPs localize in the retina of *Bbs^+/+^* mice after their topical administration. (**A**) TEM images showing the different layers of the retina of untreated *Bbs^+/+^* mice, and (**B**) the presence of the MNPs in the retina of treated *Bbs^+/+^* mice. The red arrows indicate the MNPs’ localization.

**Figure 5 pharmaceutics-13-01650-f005:**
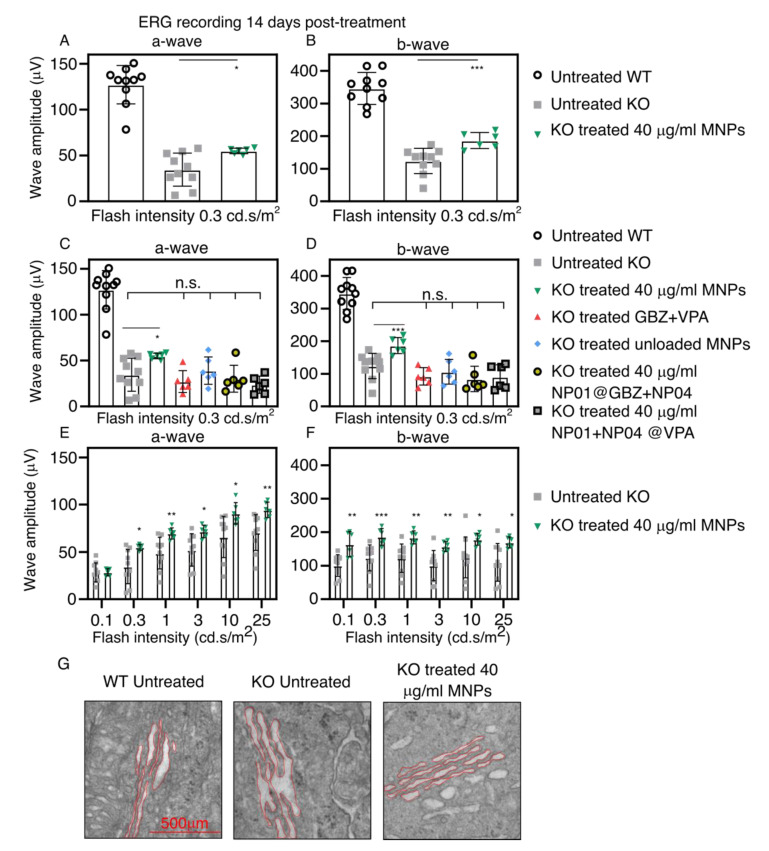
Effect of the MNP-based eye drops on the a-wave and b-wave amplitudes in the ERG measurements and on the ER dilatation. The graphics show (**A**) the average a-wave, and (**B**) the average b-wave of the untreated and treated (40 µg Fe/mL MNP with GBZ and VPA) mice at a light intensity of 0.3 cd.s/m^2^. The average a-wave (**C**) and b-wave (**D**) amplitudes recorded at a low flash intensity (0.3 cd.s/m^2^) from the untreated WT mice (white circles, black outline) and untreated KO mice (gray square) compared to the control groups: KO mice treated with GBZ and VPA but without MNPs (red triangle), unloaded MNPs (blue rhombus), MNPs loaded with either GBZ (gray circle, black outline) or VPA (gray square, black outline). The average a-wave (**E**) and b-wave (**F**) amplitudes of KO untreated mice compared to KO mice treated with MNPs (Fe 40 µg/mL loaded with GBZ and VPA) recorded in a wide range of flash intensities (0.3–25 cd.s/m^2^). (**G**) Representative TEM images showing the decrease in the ER dilatation (highlighted in red) of the mice treated with the 40 µg Fe/mL MNPs when compared to the untreated *Bbs^−/−^* mice. The mean ± SD from untreated WT mice (*n* = 9), untreated KO mice (*n* = 9), and all other groups (*n* = 6) are shown. * *p* < 0.05, ** *p* < 0.01, *** *p* < 0.005.

**Table 1 pharmaceutics-13-01650-t001:** Physicochemical properties and drug loading efficiency of the MNPs. PDI = polydispersity index. The inorganic coating corresponds to TEOS; the organic coating corresponds to citric acid; inorganic+organic corresponds to a mix of TEOS with other organosilanes.

MNPs	Drug	Coating	Zeta Potential (mV)	Hydrodynamic Size (nm)	PDI	Drug Loaded on MNPs Surface	Drug Loading Efficiency (%)
NP01	GBZ	Inorganic+Organic	−29.7 ± 9.50	141 ± 64	0.167	3.99 ± 1.99 μM	31.50 ± 15.69
NP02		Inorganic	−44.8 ± 6.6	248 ± 83	0.105	/	/
NP03		Organic	−35.0 ± 8.7	125 ± 53	0.171	3.73 ± 2.42 μM	29.49 ± 19.14
NP04	VPA	Proprietary polymer	33.0 ± 8.03	169 ± 63	0.133	1.18 ± 0.31 mM	11.81 ± 3.14
NP05		Inorganic+polymer	37.6 ± 8.93	195 ± 75	0.161	0.25 ± 0.09 mM	2.50 ± 0.79
NP06		Inorganic+organic	30.3 ± 11.1	213 ± 70	0.110	0.05 ± 0.23 mM	0.53 ± 2.35
NP07		Inorganic+organic	36.3 ± 7.60	209 ± 110	0.162	0.96 ± 0.16 mM	9.62 ± 1.60
NP08		Inorganic+organic	34.8 ± 4.74	207 ± 108	0.184	1.11 ± 0.14 mM	11.05 ± 1.39

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
