# Peer review of "Magnetically Assisted Drug Delivery of Topical Eye Drops Maintains Retinal Function In Vivo in Mice"

_pharmaceutics, 2021, doi:10.3390/pharmaceutics13101650_

Round 1

Reviewer 1 Report

This manuscript explores the idea of using topically administered magnetic nanoparticles for the delivery of small molecules to the photoreceptors of mice. The authors suggest that such an approach would be an attractive alternative to consistent intravitreal injection of such compounds, and use Bbs knockout mice as a model to demonstrate efficacy of this approach. This field is relatively new, and especially so with respect to the eye. Overall, the manuscript is well written and appears thorough. The idea is interesting, and the characterization of the nanoparticles, their in vitro stability and structure appears well done.  However, when it comes to the application of the nanoparticles on cells or in vivo, there are some improvements that need to be made, which are listed below.

Experimental:

  1. Extend the duration of toxicity experiments to longer than 24 h. A 25% reduction in viability of cultured RPE cells might biologically significant. Were significance tests used for these experiments? Why not use the combination of nanoparticles, like is done for the in vivo beneficial effect?  Possibly perform an additional, orthogonal viability assay?  Cell Titer Glo 2.0, LDH, etc?
  2. For supplemental Fig. 11, certainly perform an outlier check on 1 mg/mL and 20 ug/mL a and b waves (single mouse with v. high responses). Also, legend of this figure says 40 mg/mL and 20 mg/mL, not ug/mL
  3. Please define specifically which mice were used for the Bbs KO experiments – this is kinda buried in the supplemental – it seems like a mixture of mice may have been used according to the Methods – this is certainly not ideal… Are there differences between the ERG recordings of Bbs12-/- vs. Bbs10-/- mice?
  4. Need to explain Fig. 5 – KO treated GBZ + VPA – was this topical or intravitreal like from previous publications? Why is this not protective?
  5. Add number of TEM experiments performed in G – number of fields of view, etc. This is an important aspect because it would verify the pharmacological activity of the delivered compounds (aside from the functional ERG differences, which are surprising, to be honest).

Conceptual:

  1. VPA has lots of cellular effects, not just on the UPR. That might be the important comment with respect to this paper, but it can have a wide variety of effects mediated through HDAC inhibition.
  2. It would be good if the authors touched on the need for required re-administration of the nanoparticles – what are they envisioning? Every 2 weeks? Every month? Once? Is this
  3. Also, while the focus of this manuscript is the retina, is there a depot where the beads would accumulate elsewhere? Like in the sclera?  Would this have implications for future administration of nanoparticles?
  4. The authors note some nanoparticles in different section of the photoreceptor – but it is very difficult for the reader to compute what this means in terms of therapeutic dose and how it relates to the doses that were previously used successfully when just administering compound intravitreally.
  5. Considerations of nanoparticles and the potential for ferropoptosis, with respect specifically to the RPE? What happens to the nanoparticles eventually? Absorbed? Broken down? 

Author Response

REVIEWER #1

This manuscript explores the idea of using topically administered magnetic nanoparticles for the delivery of small molecules to the photoreceptors of mice. The authors suggest that such an approach would be an attractive alternative to consistent intravitreal injection of such compounds, and use Bbs knockout mice as a model to demonstrate efficacy of this approach. This field is relatively new, and especially so with respect to the eye. Overall, the manuscript is well written and appears thorough. The idea is interesting, and the characterization of the nanoparticles, their in vitro stability and structure appears well done.  However, when it comes to the application of the nanoparticles on cells or in vivo, there are some improvements that need to be made, which are listed below.

Experimental:

  1. Extend the duration of toxicity experiments to longer than 24 h. A 25% reduction in viability of cultured RPE cells might biologically significant. Were significance tests used for these experiments? Why not use the combination of nanoparticles, like is done for the in vivo beneficial effect?  Possibly perform an additional, orthogonal viability assay?  Cell Titer Glo 2.0, LDH, etc?

The authors thank the reviewers for this comment. The cell experiments were preliminary toxicity studies done, before in vivo experiments, to exclude the risk of severe toxicity before setting up injections in mice. In that way, 24h were sufficient to us to conclude about the absence of acute side-effect and thus start in vivo administration. By the way, these preliminary studies have been then confirmed by the absence of in vivo toxicity, confirming that no major toxicity issues exist.

Besides, the in vitro toxicity has actually been performed using the mix of particles employed in the in vivo part: in figures 2 E&F, bars marked “Mix unloaded” correspond to the mixing of the unloaded magnetic nanoparticles in 5% Glucose (i.e. without any drug) while the bars marked “Mix Loaded” are exactly the same mixture, containing valproic acid and guanabenz, that has next been topically administered to the mice. In that way, our in vitro toxicity assay excluded any severe toxicity due to MNPs and/or drugs, and can be easily correlated with the in vivo study following then.

2.  For supplemental Fig. 11, certainly perform an outlier check on 1 mg/mL and 20 ug/mL a and b waves (single mouse with v. high responses). Also, legend of this figure says 40 mg/mL and 20 mg/mL, not ug/mL

The authors thank the reviewer for pointing out the mistake in the figure. The corrections to the legends have been included in the supporting informations (Figure S11). We have run an outlier detection test but have found none. We used GraphPad and the test used was the ROUT test.

3.  Please define specifically which mice were used for the Bbs KO experiments – this is kinda buried in the supplemental – it seems like a mixture of mice may have been used according to the Methods – this is certainly not ideal… Are there differences between the ERG recordings of Bbs12-/- vs. Bbs10-/- mice?

The reviewer rises an interesting point. A previous study in our laboratory has shown that the Bbs KO models used in our experiments share similar kinetic and UPR related retinal degeneration. We clarified this point in the introduction (lines 73-74) and supported it with the relative reference [11]. This was also added in the material and methods section in lines 252-253.

4.  Need to explain Fig. 5 – KO treated GBZ + VPA – was this topical or intravitreal like from previous publications? Why is this not protective?

We thank the reviewer to rise this point. In the present study only eyedrops were applied to the animals. We did not carried out injection as they were out of scope for this study.

To clarify this point, the authors modified the title of Figure 5. Also, we added few sentences in the methods (Section 2.10, lines 280-282) that explain exactly the composition of the GBZ+VPA control.

About the lack of efficacy of topically applied GBZ+VPA without MNPs, this is the case as most small drugs molecules due to the multiple ocular  barriers (reference [1]). The we did is points to develop a system to enhance the ocular permeability, so the bioavailability, of topically applied small drug molecules.

This is well exemplified in Figure 5 C&D, where the effect of magnetic-assisted delivery was validated by including numerous controls.

5.  Add number of TEM experiments performed in G – number of fields of view, etc. This is an important aspect because it would verify the pharmacological activity of the delivered compounds (aside from the functional ERG differences, which are surprising, to be honest).

We thank the reviewer for raising this point. We have added the requested information on the corresponding materials and methods section (lines 348-351).

Conceptual:

6.  VPA has lots of cellular effects, not just on the UPR. That might be the important comment with respect to this paper, but it can have a wide variety of effects mediated through HDAC inhibition.

This is indeed an interesting point. However, in our study we did not discuss the action mechanism of VPA for several reasons. First, as stated in the introduction the treatment has already been validated for UPR-related retinal degeneration in a previous publication [12]. Second, our interest was to compare the previous use of GBZ+VPA vs magnetic-assisted, topical application of GBZ+VPA.. Searching other VPA related neuroprotective effects would not allow us to better asses the efficacy of the MNPs as drug delivery vehicle. However, studying other possible neuroprotective effects of VPA in the BBS belongs to a different study.

7.  It would be good if the authors touched on the need for required re-administration of the nanoparticles – what are they envisioning? Every 2 weeks? Every month? Once?

We thank the author to rise this point. This is probably too far fetching at this point and it would require us to speculate on this point.

The authors added a clarification on this point in the discussion in lines 673-675.

8.  Also, while the focus of this manuscript is the retina, is there a depot where the beads would accumulate elsewhere? Like in the sclera?  Would this have implications for future administration of nanoparticles?

Yes, it is possible that particle depots may be formed in other tissues. Presumably, the particles permeate to the retina via conjunctiva, sclera and choroid and some particle depots may be present in these tissues. In principle, such depots might offer further opportunities in drug delivery.

9.  The authors note some nanoparticles in different section of the photoreceptor – but it is very difficult for the reader to compute what this means in terms of therapeutic dose and how it relates to the doses that were previously used successfully when just administering compound intravitreally.

Reviewer is right. It is impossible at this point to know about quantities. It has to be noted that both our MRI and TEM experiments are only qualitative in nature as we do not comment on the concentration or quantity of particles reaching the tissue in our results. This should be admitted, but we did not claim anything about the quantity. If this is not yet in discussion, it could be added.

10.  Considerations of nanoparticles and the potential for ferropoptosis, with respect specifically to the RPE? What happens to the nanoparticles eventually? Absorbed? Broken down?

No one knows answer to this. Ferroptosis should be addressed in later studies. Studying the fate of nanoparticles in vivo is difficult. We guess all this would be relevant in the future, but not in this early POC study.

Reviewer 2 Report

In the manuscript entitled "Magnetically assisted drug delivery of topical eye drops maintains retinal function in vivo in mice", the authors tried to use magnetic nanoparticles to realize non-invasive delivery of guanabenz and valproic acid to photoreceptors in barded-Biedl syndrome mice. The following concerns should be clarified before further processing of the manuscript. 1. Figure 2E and F, the author used % for cell viability. How was the percentage calculated, with which group as the reference. The cell viability of normal cells not treated with nanoparticles should be included. Besides, some groups seems to be significantly different, but no statistical sign was included in the charts. Statistical checking should be applied to the manuscript thoroughly. 2. It would be better if invasive administration group was included as a control. 3. Figure 3, some group seems to be not significantly different. Check please. 4. The description of *, ** and *** should be included in the figure legends. 5. There are many general descriptions in the manuscript, such as very low and minor loss. This is not accurate and rigorous enough. 6. The drug release properties were not studied.

Author Response

REVIEWER #2

In the manuscript entitled "Magnetically assisted drug delivery of topical eye drops maintains retinal function in vivo in mice", the authors tried to use magnetic nanoparticles to realize non-invasive delivery of guanabenz and valproic acid to photoreceptors in barded-Biedl syndrome mice. The following concerns should be clarified before further processing of the manuscript.

  1. Figure 2E and F, the author used % for cell viability. How was the percentage calculated, with which group as the reference.

We thank the reviewer for the comment. Section 2.8 of the methods (line 250-251) and the legend of Figure 2 (line 488) have been updated accordingly.This correction has been extended to the legend of Figure S10.

The cell viability of normal cells not treated with nanoparticles should be included. Besides, some groups seems to be significantly different, but no statistical sign was included in the charts. Statistical checking should be applied to the manuscript thoroughly.

The reviewer is right, it was not specified how these percentages were obtained. The authors included the explanation in section number 2.8 of the methods (line 250-251) and the legend of Figure 2 (line 488) and in the legend of Figure S10.

The in vitro cell viability assay was used as rapid read out of NP01 and NP04 biocompatibility in retinal cells. The constituency of the observation obtained from this experiment cannot be extended to the in vivo experiment. In vitro, cells are exposed to the full dose of the MNPs and to the direct mechanical stress induced by magnetic targeting that is placed right below the cell culture place (short MNPs-magnet distance, maximal magnetic torque) whereas in vivo the MNPs are instilled on the mice eye and the magnet is placed at 1 cm from the eye (large MNP-magnet distance, low torque). Moreover, in vivo the RPE and photoreceptors cells are protected under multiple layers of tissue that hamper the direct contact of the MNPs with the cell layers of the outer retina.

For these reasons the authors believe that the information contained in Figure 2E,F is appropriate, complete and conveys adequately the conclusion that NP01 and NP04 are not toxic to RPE1 and 661W cells in the conditions that have been studied. Triplicates were carried out and the results are expressed as % using the cells treated with vehicle alone (glucose 5%) as control.

  1. It would be better if invasive administration group was included as a control.

We thank the reviewer for rising this point. However, due to short time for review (10 days) and the difficulty to get Bbs KO mice this experiment cannot be accomplished.

  1. Figure 3, some group seems to be not significantly different. Check please.

 We thank the reviewer for this comment. MRI studies are not quantitative but qualitative. In Figure 3, the images do show more MNPs-derived signal (black area, red arrow) in the area of the lens and retina when the magnet is applied compared to the condition without the magnet. This can be better observed by using the lens as reference, we see more particles close to the lens in the group treated with the magnet. Since the image represent a cut in the eye the particles that are closer to the lens would not be on the surface of the eye but inside the ocular tissues.

Moreover, this observation was validated by repeating the experiments on multiple WT mice (see Methods section 2.14, line 354).

  1. The description of *, ** and *** should be included in the figure legends.

We thank the reviewer to rise this point and we introduced modification in the manuscript (see Figure 2, line 488; Figure 5 line, 599-600; Figure S10 and Figure S11).

  1. There are many general descriptions in the manuscript, such as very low and minor loss. This is not accurate and rigorous enough.

We thank the reviewer to rise this point and we reviewed both the manuscript  and supplementary information.

      6.  The drug release properties were not studied.

We thank the reviewer for rising this point. Release studies were carried out for both particles, but they were not conclusive because of some technical difficulties that have been disclosed in the manuscript (Line 644-661) and supported with references on the topic [43,44,45,46].  

Below the reviewer will find the technical details of the drug release study.

  1. A first release study was carried out by placing 1 ml of the loaded MNPs in a dialysis bag that was placed in a test tube containing 10 ml of release medium (without drug) at 37°C. Sink conditions were applied: the total amount of release medium was replaced at fixed intervals of time, faster at the beginning (first 10 minutes) and slower in the second part of the study. Special attention was given to counterbalance the effect of evaporation in the second part of the study. Then, the concentration of the drug contained in each fraction of release medium was measured by HPLC.

 This experiment failed in recovering 100% of the initial concentration of the drugs. The explanation given was that, as from Table 1, GBZ and VPA loading on the surface of NP01 and NP04, respectively, was 3.99±1.99 µM and 1.18±0.31 mM. These concentrations were already in the lowest part of the limit of detection of our HPLC system (Figure S1) (see section 2.6 of the methods for the detailed description of how these concentrations were measured).

Because of the additional dilution factor of the release study we could not detect most of the drugs. 

  1. A second release study was carried out following the procedure of the first one with the difference that all the fractions of the released medium collected during the study were freeze-dried. Next, the test tubes were hydrated using 1 ml of pure MilliQ water prior HPLC analysis. This procedure would have allowed to obtain concentrations 10 times higher compared to the previous release study. 

Also, two new calibration curves were established for the HPLC, one for GBZ and one for VPA, to take in consideration the higher concentration of the components of the release buffer (now 10 times more concentrated).

 This experiment failed to recover the initial amount of drug. GBZ is very difficult to dissolve in pure water while VPA is little easier. The fact of having 10 times more concentrated components of the release medium may have hampered the appropriate dissolution of GBZ and VPA, so they could not be detected in sufficient amount with our HPLC system.

  1. A third attempt was carried out just for VPA. This time each fraction collected of the release medium was first acidified with 1M HCl and then extracted three times with HPLC grade dichloromethane. This procedure would have allowed to leave behind the majority of the of inorganic salts. Later, the organic solvent was evaporated and the residual was rehydrated with 1 ml of MilliQ water. However, this experiment failed to recover the initial amount of VPA.

Due to these technical difficulties and because of the large amount of time taken by HPLC analysis, this experiment was suspended.

Yet, it is very important to underline here that the loading of GBZ and VPA on NP01 and NP04, respectively, was mediated only by electrostatic interaction. Thus, classic burst release effect must be expected for these two delivery systems, hence leading to the separation of the drug from the MNPs surface in short time (minutes).  

Moreover, this consideration leads to another very important point that must be highlighted here. MNPs do not improve the delivery of drugs because they penetrate more the tissues under the effect of a pulling magnetic force. The uptake of MNPs by cells is governed only by endocytosis reference [14].

MNPs are used to overcome diffusion related processes. In combination with magnetic targeting, MNPs allow to sediment and focus a large part of the therapeutic dose on the target within few minutes (fast kinetics). This in turn, boosts the tissue/cellular penetration of drugs. In our manuscript, we have demonstrated that this advantage of MNPs over other delivery systems (i.e. liposomes, reference [40]) boosts further the uptake of small drug molecules into the eye, presumably from the conjunctiva.

Round 2

Reviewer 2 Report

The author appropriately responded to the concerns.